# Truncated Horizon Policy Search: Combining Reinforcement Learning & Imitation Learning

**Wen Sun**
Robotics Institute
Carnegie Mellon University
Pittsburgh, PA, USA
`wensun@cs.cmu.edu`

**J. Andrew Bagnell**
Robotics Institute
Carnegie Mellon University
Pittsburgh, PA, USA
`dbagnell@cs.cmu.edu`

**Byron Boots**
School of Interactive Computing
Georgia Institute of Technology
Atlanta, GA, USA
`bboots@cc.gatech.edu`

## Abstract

In this paper, we propose to combine imitation and reinforcement learning via the idea of reward shaping using an oracle. We study the effectiveness of the near-optimal cost-to-go oracle on the planning horizon and demonstrate that the cost-to-go oracle shortens the learner's planning horizon as function of its accuracy: a globally optimal oracle can shorten the planning horizon to one, leading to a one-step greedy Markov Decision Process which is much easier to optimize, while an oracle that is far away from the optimality requires planning over a longer horizon to achieve near-optimal performance. Hence our new insight bridges the gap and interpolates between imitation learning and reinforcement learning. Motivated by the above mentioned insights, we propose Truncated HORizon Policy Search (THOR), a method that focuses on searching for policies that maximize the total reshaped reward over a finite planning horizon when the oracle is sub-optimal. We experimentally demonstrate that a gradient-based implementation of THOR can achieve superior performance compared to RL baselines and IL baselines even when the oracle is sub-optimal.

## 1 Introduction

Reinforcement Learning (RL), equipped with modern deep learning techniques, has dramatically advanced the state-of-the-art in challenging sequential decision problems including high-dimensional robotics control tasks as well as video and board games (Mnih et al., 2015; Silver et al., 2016). However, these approaches typically require a large amount of training data and computational resources to succeed. In response to these challenges, researchers have explored strategies for making RL more efficient by leveraging additional information to guide the learning process. Imitation learning (IL) is one such approach. In IL, the learner can reference expert demonstrations (Abbeel & Ng, 2004), or can access a cost-to-go oracle (Ross & Bagnell, 2014), providing additional information about the long-term effects of learner decisions. Through these strategies, imitation learning lowers sample complexity by reducing random global exploration. For example, Sun et al. (2017) shows that, with access to an optimal expert, imitation learning can exponentially lower sample complexity compared to pure RL approaches. Experimentally, researchers also have demonstrated sample efficiency by leveraging expert demonstrations by adding demonstrations into a replay buffer (Večerík et al., 2017; Nair et al., 2017), or mixing the policy gradient with a behavioral cloning-related gradient (Rajeswaran et al., 2017).

Although imitating experts *can* speed up the learning process in RL tasks, the performance of the learned policies are generally limited to the performance of the expert, which is often sub-optimal in practice. Previous imitation learning approaches with strong theoretical guarantees such as Data Aggregation (DAgger) (Ross et al., 2011) and Aggregation with Values (AggreVaTe) (Ross & Bagnell, 2014) can only guarantee a policy which performs as well as the expert policy or a one-step deviation improvement over the expert policy.[1] Unfortunately, this implies that imitation learning with a sub-optimal expert will often return a sub-optimal policy. Ideally, we want the best of both

---

[1] AggreVaTe achieves one-step deviation improvement over the expert under the assumption that the policy class is rich enough.

IL and RL: we want to use the expert to quickly learn a reasonable policy by imitation, while also exploring how to improve upon the expert with RL. This would allow the learner to overcome the sample inefficiencies inherent in a pure RL strategy while also allowing the learner to eventually surpass a potentially sub-optimal expert. Combining RL and IL is, in fact, not new. Chang et al. (2015) attempted to combine IL and RL by stochastically interleaving incremental RL and IL updates. By doing so, the learned policy will either perform as well as the expert policy–the property of IL (Ross & Bagnell, 2014), or eventually reach a local optimal policy–the property of policy iteration-based RL approaches. Although, when the expert policy is sub-optimal, the learned locally optimal policy could potentially perform better than the expert policy, it is still difficult to precisely quantify how much the learner can improve over the expert.

In this work, we propose a novel way of combining IL and RL through the idea of *Reward Shaping* (Ng et al., 1999). Throughout our paper we use *cost* instead of reward, and we refer to the concept of reward shaping with costs as *cost shaping*. We assume access to a cost-to-go oracle that provides an estimate of expert cost-to-go during training. The key idea is that the cost-to-go oracle can serve as a potential function for cost shaping. For example, consider a task modeled by a *Markov Decision Process* (MDP). Cost shaping with the cost-to-go oracle produces a new MDP with an optimal policy that is equivalent to the optimal policy of the original MDP (Ng et al., 1999). The idea of cost shaping naturally suggests a strategy for IL: pick a favourite RL algorithm and run it on the new MDP reshaped using expert's cost-to-go oracle. In fact, Ng et al. (1999) demonstrated that running SARSA (Sutton & Barto, 1998) on an MDP reshaped with a potential function that approximates the optimal policy's value-to-go, is an effective strategy.

We take this idea one step further and study the effectiveness of the cost shaping with the expert's cost-to-go oracle, with a focus on the setting where we only have an *imperfect* estimator $\hat{V}^e$ of the cost-to-go of some expert policy $\pi^e$, i.e., $\hat{V}^e \neq V^*$, where $V^*$ is the optimal policy's cost-to-go in the original MDP. We show that cost shaping with the cost-to-go oracle shortens the learner's planning horizon as a function of the accuracy of the oracle $\hat{V}^e$ compared to $V^*$. Consider two extremes. On one hand, when we reshape the cost of the original MDP with $V^*$ (i.e., $\hat{V}^e = V^*$), the reshaped MDP has an effective planning horizon of one: a policy that minimizes the one-step cost of the reshaped MDP is in fact the optimal policy (hence the optimal policy of the original MDP). On the other hand, when the cost-to-go oracle provides no information regarding $V^*$, we have no choice but simply optimize the reshaped MDP (or just the original MDP) using RL over the entire planning horizon.

With the above insight, we propose the high-level strategy for combining IL and RL, which we name *Truncated HORizon Policy Search with cost-to-go oracle* (THOR). The idea is to first shape the cost using the expert's cost-to-go oracle $\hat{V}^e$, and then truncate the planning horizon of the new MDP and search for a policy that optimizes over the truncated planning horizon. For discrete MDPs, we mathematically formulate this strategy and guarantee that we will find a policy that performs better than the expert with a gap that can be exactly quantified (which is missing in the previous work of Chang et al. (2015)). In practice, we propose a gradient-based algorithm that is motivated from this insight. The practical algorithm allows us to leverage complex function approximators to represent policies and can be applied to continuous state and action spaces. We verify our approach on several MDPs with continuous state and action spaces and show that THOR can be much more sample efficient than strong RL baselines (we compared to *Trust Region Policy Optimization with Generalized Advantage Estimation (TRPO-GAE)* (Schulman et al., 2016)), and can learn a significantly better policy than AGGREVATE (we compared to the policy gradient version of AGGREVATE from (Sun et al., 2017)) with access only to an imperfect cost-to-go oracle.

## 1.1 RELATED WORK AND OUR CONTRIBUTION

Previous work has shown that truncating the planning horizon can result in a tradeoff between accuracy and computational complexity. Farahmand et al. (2016) proposed a model-based RL approach that focuses on a search for policies that maximize a sum of $k$-step rewards with a termination value that approximates the optimal value-to-go. Their algorithm focuses on the model-based setting and the discrete state and action setting, as the algorithm needs to perform $k$-step value iteration to compute the policy. Another use of the truncated planning horizon is to trade off bias and variance. When the oracle is an approximation of the value function of the agent's current policy, by using k-

step rollouts bottomed up by the oracle's return, truncating the planning horizon trades off bias and variance of the estimated reward-to-go. The bias-variance tradeoff has been extensively studied in Temporal Difference Learning literature (Sutton, 1988) and policy iteration literature as well (Gabillon et al., 2011). Ng (2003) is perhaps the closest to our work. In Theorem 5 in the Appendix of Ng's dissertation, Ng considers the setting where the potential function for reward shaping is close to the optimal value function and suggests that if one performs reward shaping with the potential function, then one can decrease the discount factor of the original MDP without losing the optimality that much. Although in this work we consider truncating the planning steps directly, Theorem 5 in Ng's dissertation and our work both essentially considers trading off between the hardness of the reshaped MDP (the shorter the planning horizon, the easier the MDP to optimize) and optimality of the learned policy. In addition to this tradeoff, our work suggests a path toward understanding previous imitation learning approaches through reward shaping, and tries to unify IL and RL by varying the planning horizon from 1 to infinity, based on how close the expert oracle is to the optimal value function. Another contribution of our work is a lower bound analysis that shows that performance limitation of AGGREVATE with an imperfect oracle, which is missing in previous work (Ross & Bagnell, 2014). The last contribution of our work is a model-free, actor-critic style algorithm that can be used for continuous state and action spaces.

## 2 PRELIMINARIES

We consider the problem of optimizing Markov Decision Process defined as $\mathcal{M}_0 = (\mathcal{S}, \mathcal{A}, P, C, \gamma)$. Here, $\mathcal{S}$ is a set of $S$ states and $\mathcal{A}$ is a set of $A$ actions; $P$ is the transition dynamics at such that for any $s \in \mathcal{S}, s' \in \mathcal{S}, a \in \mathcal{A}$, $P(s'|s, a)$ is the probability of transitioning to state $s'$ from state $s$ by taking action $a$. For notation simplicity, in the rest of the paper, we will use short notation $P_{sa}$ to represent the distribution $P(\cdot|s, a)$. The cost for a given pair of $s$ and $a$ is $c(s, a)$, which is sampled from the cost distribution $C(s, a)$ with mean value $\bar{c}(s, a)$. A stationary stochastic policy $\pi(a|s)$ computes the probability of generating action $a$ given state $s$.

The value function $V_{\mathcal{M}_0}^\pi$ and the state action cost-to-go $Q_{\mathcal{M}_0, h}^\pi(s, a)$ of $\pi$ on $\mathcal{M}_0$ are defined as:

$$V_{\mathcal{M}_0}^\pi(s) = \mathbb{E}\Big[\sum_{t=0}^\infty \gamma^t c(s_t, a_t)|s_0 = s, a \sim \pi\Big], \quad Q_{\mathcal{M}_0}^\pi(s, a) = \mathbb{E}\big[c(s, a) + \gamma \mathbb{E}_{s' \sim P_{sa}}[V_{\mathcal{M}_0}^\pi(s')]\big],$$

where the expectation is taken with respect to the randomness of $\mathcal{M}_0$ and the stochastic policy $\pi$. With $V_{\mathcal{M}_0}^\pi$ and $Q_{\mathcal{M}_0}^\pi$, we define the disadvantage[2] function $A_{\mathcal{M}_0}^\pi(s, a) = Q_{\mathcal{M}_0}^\pi(s, a) - V_{\mathcal{M}_0}^\pi(s)$. The objective is to search for the optimal policy $\pi^*$ such that $\pi^* = \arg\min_\pi V^\pi(s), \forall s \in \mathcal{S}$.

Throughout this work, we assume access to an cost-to-go oracle $\hat{V}^e(s) : \mathcal{S} \to \mathbb{R}$. Note that we do not require $\hat{V}^e(s)$ to be equal to $V_{\mathcal{M}_0}^*$. For example, $\hat{V}^e(s)$ could be obtained by learning from trajectories demonstrated by the expert $\pi^e$ (e.g., Temporal Difference Learning (Sutton & Barto, 1998)), or $\hat{V}^e$ could be computed by near-optimal search algorithms via access to ground truth information (Daumé III et al., 2009; Chang et al., 2015; Sun et al., 2016) or via access to a simulator using Dynamic Programming (DP) techniques (Choudhury et al., 2017; Pan et al., 2017). In our experiment, we focus on the setting where we learn a $\hat{V}^e(s)$ using TD methods from a set of expert demonstrations.

### 2.1 COST SHAPING

Given the original MDP $\mathcal{M}_0$ and any potential functions $\Phi : \mathcal{S} \to \mathbb{R}$, we can reshape the cost $c(s, a)$ sampled from $C(s, a)$ to be:

$$c'(s, a) = c(s, a) + \gamma \Phi(s') - \Phi(s), \quad s' \sim P_{sa}. \tag{1}$$

Denote the new MDP $\mathcal{M}$ as the MDP obtained by replacing $c$ by $c'$ in $\mathcal{M}_0$: $\mathcal{M} = (\mathcal{S}, \mathcal{A}, P, c', \gamma)$. Ng et al. (1999) showed that the optimal policy $\pi_{\mathcal{M}}^*$ on $\mathcal{M}$ and the optimal policy $\pi_{\mathcal{M}_0}^*$ on the original MDP are the same: $\pi_{\mathcal{M}}^*(s) = \pi_{\mathcal{M}_0}^*(s), \forall s$. In other words, if we can successfully find $\pi_{\mathcal{M}}^*$ on $\mathcal{M}$, then we also find $\pi_{\mathcal{M}_0}^*$, the optimal policy on the original MDP $\mathcal{M}_0$ that we ultimately want to optimize.

---

[2] We call $A^\pi$ as the *disadvantage* function as we are working in the cost setting.

## 2.2 IMITATION LEARNING

In IL, when given a cost-to-go oracle $V^e$, we can use it as a potential function for cost shaping. Specifically let us define the disadvantage $A^e(s, a) = c(s, a) + \gamma \mathbb{E}_{s' \sim P_{sa}}[V^e(s')] - V^e(s)$. As cost shaping does not change the optimal policy, we can rephrase the original policy search problem using the shaped cost:

$$\pi^* = \arg\min_\pi \mathbb{E}[\sum_{t=0}^{\infty} \gamma^t A^e(s_t, a_t)|s_0 = s, a \sim \pi], \tag{2}$$

for all $s \in \mathcal{S}$. Though Eq. 2 provides an alternative objective for policy search, it could be as hard as the original problem as $\mathbb{E}[\sum_t \gamma^t A^e(s_t, a_t)]$ is just equal to $\mathbb{E}[\sum_t \gamma^t c(s_t, a_t) - V^e(s_0)]$, which can be easily verified using the definition of cost shaping and a telescoping sum trick.

As directly optimizing Eq 2 is as difficult as policy search in the original MDP, previous IL algorithms such as AGGREVATE essentially ignore temporal correlations between states and actions along the planning horizon and directly perform a policy iteration over the expert policy at every state, i.e., they are greedy with respect to $A^e$ as $\hat{\pi}(s) = \arg\min_a A^e(s, a), \forall s \in \mathcal{S}$. The policy iteration theorem guarantees that such a greedy policy $\hat{\pi}$ performs at least as well as the expert. Hence, when the expert is optimal, the greedy policy $\hat{\pi}$ is guaranteed to be optimal. However when $V^e$ is not the optimal value function, the greedy policy $\hat{\pi}$ over $A^e$ is a one-step deviation improvement over the expert but is not guaranteed to be close to the optimal $\pi^*$. We analyze in detail how poor the policy resulting from such a greedy policy improvement method could be when $V^e$ is far away from the optimal value function in Sec. 3.

## 3 EFFECTIVENESS OF COST-TO-GO ORACLE ON PLANNING HORIZON

In this section we study the dependency of effective planning horizon on the cost-to-go oracle. We focus on the setting where we have access to an oracle $\hat{V}^e(s)$ which approximates the cost-to-go of some expert policy $\pi^e$ (e.g., $V^e$ could be designed by domain knowledge (Ng et al., 1999) or learned from a set of expert demonstrations). We assume the oracle is close to $V^*_{\mathcal{M}_0}$, but imperfect: $|\hat{V}^e - V^*_{\mathcal{M}_0}| = \epsilon$ for some $\epsilon \in \mathbb{R}^+$. We first show that with such an imperfect oracle, previous IL algorithms AGGREVATE and AGGREVATE D (Ross & Bagnell, 2014; Sun et al., 2017) are only guaranteed to learn a policy that is $\gamma\epsilon/(1-\gamma)$ away from the optimal. Let us define the expected total cost for any policy $\pi$ as $J(\pi) = \mathbb{E}_{s_0 \sim v}[V^\pi_{\mathcal{M}_0}(s_0)]$, measured under some initial state distribution $v$ and the original MDP $\mathcal{M}_0$.

**Theorem 3.1.** *There exists an MDP and an imperfect oracle $\hat{V}^e(s)$ with $|\hat{V}^e(s) - V^*_{\mathcal{M}_0,h}(s)| = \epsilon$, such that the performance of the induced policy from the cost-to-go oracle $\hat{\pi}^* = \arg\min_a \left[c(s, a) + \gamma\mathbb{E}_{s' \sim P_{sa}}[\hat{V}^e(s')]\right]$ is at least $\Omega(\gamma\epsilon/(1 - \gamma))$ away from the optimal policy $\pi^*$:*

$$J(\hat{\pi}^*) - J(\pi^*) \geq \Omega\left(\frac{\gamma}{1 - \gamma}\epsilon\right). \tag{3}$$

The proof with the constructed example can be found in Appendix A. Denote $\hat{Q}^e(s, a) = c(s, a) + \gamma\mathbb{E}_{s'}[\hat{V}^e(s')]$, in high level, we construct an example where $\hat{Q}^e$ is close to $Q^*$ in terms of $\|\hat{Q}^e - Q^*\|_\infty$, but the order of the actions induced by $\hat{Q}^e$ is different from the order of the actions from $Q^*$, hence forcing the induced policy $\hat{\pi}^*$ to make mistakes.

As AGGREVATE at best computes a policy that is one-step improvement over the oracle, i.e., $\hat{\pi}^* = \arg\min_a \left[c(s, a) + \gamma\mathbb{E}_{s' \sim P_{sa}}[\hat{V}^e(s')]\right]$, it eventually has to suffer from the above lower bound. This $\epsilon$ gap in fact is not surprising as AGGREVATE is a *one-step greedy* algorithm in a sense that it is only optimizing the one-step cost function $c'$ from the *reshaped* MDP $\mathcal{M}$. To see this, note that the cost of the reshaped MDP $\mathcal{M}$ is $\mathbb{E}[c'(s, a)] = [c(s, a) + \gamma\mathbb{E}_{s' \sim P_{sa}}\hat{V}^e(s') - \hat{V}^e(s)]$, and we have $\hat{\pi}^*(s) = \arg\min_a \mathbb{E}[c'(s, a)]$. Hence AGGREVATE can be regarded as a special algorithm that aims to optimizing the one-step cost of MDP $\mathcal{M}$ that is reshaped from the original MDP $\mathcal{M}_0$ using the cost-to-go oracle.

Though when the cost-to-go oracle is imperfect, AGGREVATE will suffer from the above lower bound due to being greedy, when the cost-to-go oracle is perfect, i.e., $\hat{V}^e = V^*$, being greedy on one-step cost makes perfect sense. To see this, use the property of the cost shaping (Ng et al., 1999), we can verify that when $\hat{V}^e = V^*$:

$$V_{\mathcal{M}}^*(s) = 0, \quad \pi_{\mathcal{M}}^*(s) = \arg\min_a \mathbb{E}[c'(s,a)], \quad \forall s \in \mathcal{S}. \tag{4}$$

Namely the optimal policy on the reshaped MDP $\mathcal{M}$ only optimizes the one-step cost, which indicates that the optimal cost-to-go oracle shortens the planning horizon to one: finding the optimal policy on $\mathcal{M}_0$ becomes equivalent to optimizing the immediate cost function on $\mathcal{M}$ at every state $s$.

When the cost-to-go oracle is $\epsilon$ away from the optimality, we lose the one-step greedy property shown in Eq. 4. In the next section, we show that how we can break the lower bound $\Omega(\epsilon/(1-\gamma))$ only with access to an imperfect cost-to-go oracle $\hat{V}^e$, by being less greedy and looking head for more than one-step.

### 3.1 OUTPERFORMING THE EXPERT

Given the reshaped MDP $\mathcal{M}$ with $\hat{V}^e$ as the potential function, as we mentioned in Sec. 2.2, directly optimizing Eq. 2 is as difficult as the original policy search problem, we instead propose to minimize the total cost of a policy $\pi$ over a finite $k \geq 1$ steps at any state $s \in \mathcal{S}$:

$$\mathbb{E}\Big[ \sum_{i=1}^{k} \gamma^{i-1} c'(s_i, a_i) | s_1 = s; a \sim \pi \Big]. \tag{5}$$

Using the definition of cost shaping and telescoping sum trick, we can re-write Eq. 5 in the following format, which we define as $k$-step disadvantage with respect to the cost-to-go oracle:

$$\mathbb{E}\Big[ \sum_{i=1}^{k} \gamma^{i-1} c(s_i, a_i) + \gamma^k \hat{V}^e(s_{k+1}) - \hat{V}^e(s_1) | s_1 = s; a \sim \pi \Big], \forall s \in \mathcal{S}. \tag{6}$$

We assume that our policy class $\Pi$ is rich enough that there always exists a policy $\hat{\pi}^* \in \Pi$ that can simultaneously minimizes the $k-$step disadvantage at every state (e.g., policies in tabular representation in discrete MDPs). Note that when $k = 1$, minimizing Eq. 6 becomes the problem of finding a policy that minimizes the disadvantage $A_{\mathcal{M}_0}^e(s,a)$ with respect to the expert and reveals AGGREVATE.

The following theorem shows that to outperform expert, we can optimize Eq. 6 with $k > 1$. Let us denote the policy that minimizes Eq. 6 in every state as $\hat{\pi}^*$, and the value function of $\hat{\pi}^*$ as $V^{\hat{\pi}^*}$.

**Theorem 3.2.** *Assume $\hat{\pi}^*$ minimizes Eq. 6 for every state $s \in \mathcal{S}$ with $k > 1$ and $|\hat{V}^e(s) - V^*(s)| = \Theta(\epsilon), \forall s$. We have :*

$$J(\hat{\pi}^*) - J(\pi^*) \leq O\left( \frac{\gamma^k}{1 - \gamma^k} \epsilon \right) \tag{7}$$

Compare the above theorem to the lower bound shown in Theorem 3.1, we can see that when $k > 1$, we are able to learn a policy that performs better than the policy induced by the oracle (i.e., $\hat{\pi}^*(s) = \arg\min_a \hat{Q}^e(s,a)$) by at least $(\frac{\gamma}{1-\gamma} - \frac{\gamma^k}{1-\gamma^k})\epsilon$. The proof can be found in Appendix B.

Theorem 3.2 and Theorem 3.1 together summarize that when the expert is imperfect, simply computing a policy that minimizes the one-step disadvantage (i.e., $(k = 1)$) is not sufficient to guarantee near-optimal performance; however, optimizing a $k$-step disadvantage with $k > 1$ leads to a policy that guarantees to *outperform the policy induced by the oracle* (i.e., the best possible policy that can be learnt using AGGREVATE and AGGREVATED). Also our theorem provides a concrete performance gap between the policy that optimizes Eq. 6 for $k > 1$ and the policy that induced by the oracle, which is missing in previous work (e.g., (Chang et al., 2015)).

As we already showed, if we set $k = 1$, then optimizing Eq. 6 becomes optimizing the disadvantage over the expert $A_{\mathcal{M}_0}^e$, which is exactly what AGGREVATE aims for. When we set $k = \infty$, optimizing Eq. 6 or Eq. 5 just becomes optimizing the total cost of the original MDP. Optimizing over a shorter

---

**Algorithm 1** `Truncated Horizon Policy Search (THOR)`

---

1: **Input:** The original MDP $\mathcal{M}_0$. Truncation Step $k$. Oracle $V^e$.
2: Initialize policy $\pi_{\theta_0}$ with parameter $\theta_0$ and truncated advantage estimator $\hat{A}_{\mathcal{M}}^{0,k}$.
3: **for** n = 0, ... **do**
4:     Reset system.
5:     Execute $\pi_{\theta_n}$ to generate a set of trajectories $\{\tau_i\}_{i=1}^N$.
6:     Reshape cost $c'(s_t, a_t) = c(s_t, a_t) + V_{t+1}^e(s_{t+1}) - V_t^e(s_t)$, for every $t \in [1, |\tau_i|]$ in every trajectory $\tau_i, i \in [N]$.
7:     Compute gradient:

$$\sum_{\tau_i} \sum_t \nabla_\theta (\ln \pi_\theta(a_t|s_t))|_{\theta=\theta_n} \hat{A}_{\mathcal{M}}^{\pi_n,k}(s_t, a_t) \tag{8}$$

8:     Update disadvantage estimator to $\hat{A}_{\mathcal{M}}^{\pi_n,k}$ using $\{\tau_i\}_i$ with reshaped cost $c'$.
9:     Update policy parameter to $\theta_{n+1}$.
10: **end for**

---

finite horizon is easier than optimizing over the entire infinite long horizon due to advantages such as smaller variance of the empirical estimation of the objective function, less temporal correlations between states and costs along a shorter trajectory. Hence our main theorem essentially provides a tradeoff between the optimality of the solution $\hat{\pi}^*$ and the difficulty of the underlying optimization problem.

## 4 Practical Algorithm

Given the original MDP $\mathcal{M}_0$ and the cost-to-go oracle $\hat{V}^e$, the reshaped MDP's cost function $c'$ is obtained from Eq. 1 using the cost-to-go oracle as a potential function. Instead of directly applying RL algorithms on $\mathcal{M}_0$, we use the fact that the cost-to-go oracle shortens the effective planning horizon of $\mathcal{M}$, and propose *THOR: Truncated HORizon Policy Search* summarized in Alg. 1. The general idea of THOR is that instead of searching for policies that optimize the total cost over the entire infinitely long horizon, we focus on searching for polices that minimizes the total cost over a truncated horizon, i.e., a $k-$step time window. Below we first show how we derive THOR from the insight we obtained in Sec. 3.

Let us define a $k$-step truncated value function $V_{\mathcal{M}}^{\pi,k}$ and similar state action value function $Q_{\mathcal{M}}^{\pi,k}$ on the reshaped MDP $\mathcal{M}$ as:

$$V_{\mathcal{M}}^{\pi,k}(s) = \mathbb{E}\Big[\sum_{t=1}^k \gamma^{t-1} c'(s_t, a_t)|s_1 = s, a \sim \pi\Big],$$

$$Q_{\mathcal{M}}^{\pi,k}(s,a) = \mathbb{E}\Big[c'(s,a) + \sum_{i=1}^{k-1} \gamma^i c'(s_i, a_i)|s_i \sim P_{sa}, a_i \sim \pi(s_i)\Big], \tag{9}$$

At any time state $s$, $V_{\mathcal{M}}^{\pi,k}$ only considers (reshaped) cost signals $c'$ from a k-step time window.

We are interested in searching for a policy that can optimizes the total cost over a finite k-step horizon as shown in Eq. 5. For MDPs with large or continuous state spaces, we cannot afford to enumerate all states $s \in \mathcal{S}$ to find a policy that minimizes the $k-$step disadvantage function as in Eq. 5. Instead one can leverage the approximate policy iteration idea and minimize the weighted cost over state space using a state distribution $\nu$ (Kakade & Langford, 2002; Bagnell et al., 2004):

$$\min_{\pi \in \Pi} \mathbb{E}_{s_0 \sim \nu} \left[\mathbb{E}\Big[\sum_{i=1}^k \gamma^i c'(s_i, a_i)|a \sim \pi\Big]\right]. \tag{10}$$

For parameterized policy $\pi$ (e.g., neural network policies), we can implement the minimization in Eq. 10 using gradient-based update procedures (e.g., Stochastic Gradient Descent, Natural Gradient (Kakade, 2002; Bagnell & Schneider, 2003)) in the policy's parameter space. In the setting where

the system cannot be reset to any state, a typical choice of exploration policy is the currently learned policy (possibly mixed with a random process (Lillicrap et al., 2015) to futher encourage exploration). Denote $\pi_n$ as the currently learned policy after iteration $n$ and $Pr_{\pi_n}(\cdot)$ as the average state distribution induced by executing $\pi_n$ (parameterized by $\theta_n$) on the MDP. Replacing the exploration distribution by $Pr_{\pi_n}(\cdot)$ in Eq. 10, and taking the derivative with respect to the policy parameter $\theta$, the policy gradient is:

$$\mathbb{E}_{s\sim Pr_{\pi_n}}\left[\mathbb{E}_{\tau^k\sim\pi_n}[\sum_{i=1}^{k}\nabla_\theta\ln\pi(a_i|s_i;\theta)(\sum_{j=i}^{k+i}\gamma^{j-i}c'(s_j,a_j))]\right]$$

$$\approx\mathbb{E}_{s\sim Pr_{\pi_n}}\left[\mathbb{E}_{\tau^k\sim\pi_n}[\sum_{i=1}^{k}\nabla_\theta\ln\pi(a_i|s_i;\theta)Q_\mathcal{M}^{\pi,k}(s_i,a_i)]\right]$$

where $\tau^k\sim\pi_n$ denotes a partial $k-$step trajectory $\tau^k=\{s_1,a_1,...,s_k,a_k|s_1=s\}$ sampled from executing $\pi_n$ on the MDP from state $s$. Replacing the expectation by empirical samples from $\pi_n$, replacing $Q_\mathcal{M}^{\pi,k}$ by a critic approximated by Generalized disadvantage Estimator (GAE) $\hat{A}_\mathcal{M}^{\pi,k}$ (Schulman et al., 2016), we get back to the gradient used in Alg. 1:

$$\mathbb{E}_{s\sim Pr_{\pi_n}}\left[\mathbb{E}_{\tau_k\sim\pi_n}[\sum_{i=1}^{k}\nabla_\theta\ln\pi(a_i|s_i;\theta)Q_\mathcal{M}^{\pi,k}(s_i,a_i)]\right]$$

$$\approx k\sum_\tau\left(\sum_{t=1}^{|\tau|}\nabla_\theta\ln(\pi(a_t|s_t;\theta))\hat{A}_\mathcal{M}^{\pi,k}(s_t,a_t)\right)/H, \tag{11}$$

where $|\tau|$ denotes the length of the trajectory $\tau$.

## 4.1 Interpretation using Truncated Back-Propagation Through Time

If using the classic policy gradient formulation on the reshaped MDP $\mathcal{M}$ we should have the following expression, which is just a re-formulation of the classic policy gradient (Williams, 1992):

$$\mathbb{E}_\tau\sum_{t=1}^{|\tau|}\left(c_t'\sum_{i=0}^{t-1}(\nabla_\theta\ln\pi(a_{t-i}|s_{t-i};\theta))\right), \tag{12}$$

which is true since the cost $c_i'$ (we denote $c_i'(s,a)$ as $c_i'$ for notation simplicity) at time step $i$ is correlated with the actions at time step $t=i$ all the way back to the beginning $t=1$. In other words, in the policy gradient format, the effectiveness of the cost $c_t$ is *back-propagated through time* all the way back the first step. Our proposed gradient formulation in Alg. 1 shares a similar spirit of *Truncated Back-Propagation Through Time* (Zipser, 1990), and can be regarded as a truncated version of the classic policy gradient formulation: at any time step $t$, the cost $c'$ is back-propagated through time at most $k$-steps:

$$\mathbb{E}_\tau\sum_{t=1}^{|\tau|}\left(c_t'\sum_{i=0}^{k-1}(\nabla_\theta\ln\pi(a_{t-i}|s_{t-i};\theta))\right), \tag{13}$$

In Eq. 13, for any time step $t$, we ignore the correlation between $c_t'$ and the actions that are executed $k$-step before $t$, hence eliminates long temporal correlations between costs and old actions. In fact, AGGREVATE D (Sun et al., 2017), a policy gradient version of AGGREVATE, sets $k=1$ and can be regarded as *No Back-Propagation Through Time*.

## 4.2 CONNECTION TO IL AND RL

The above gradient formulation provides a natural half-way point between IL and RL. When $k = 1$ and $\hat{V}^e = V_{\mathcal{M}_0}^*$ (the optimal value function in the original MDP $\mathcal{M}_0$):

$$\mathbb{E}_\tau \Big[ \sum_{t=1}^{|\tau|} \nabla_\theta (\ln \pi_\theta(a_t|s_t)) A_{\mathcal{M}}^{\pi^e,1}(s_t, a_t) \Big] = \mathbb{E}_\tau \Big[ \sum_{t=1}^{|\tau|} \nabla_\theta (\ln \pi_\theta(a_t|s_t)) Q_{\mathcal{M}}^{\pi_e,1}(s_t, a_t) \Big]$$

$$= \mathbb{E}_\tau \Big[ \sum_{t=1}^{|\tau|} \nabla_\theta (\ln \pi_\theta(a_t|s_t)) \mathbb{E}[c'(s_t, a_t)] \Big] = \mathbb{E}_\tau \Big[ \sum_{t=1}^{|\tau|} \nabla_\theta (\ln \pi_\theta(a_t|s_t)) A_{\mathcal{M}_0}^{\pi^*}(s_t, a_t) \Big], \quad (14)$$

where, for notation simplicity, we here use $\mathbb{E}_\tau$ to represent the expectation over trajectories sampled from executing policy $\pi_\theta$, and $A_{\mathcal{M}_0}^\pi$ is the advantage function on the original MDP $\mathcal{M}_0$. The fourth expression in the above equation is exactly the gradient proposed by AGGREVATED (Sun et al., 2017). AGGREVATED performs gradient descent with gradient in the format of the fourth expression in Eq. 14 to discourage the log-likelihood of an action $a_t$ that has low advantage over $\pi^*$ at a given state $s_t$.

On the other hand, when we set $k = \infty$, i.e., no truncation on horizon, then we return back to the classic policy gradient on the MDP $\mathcal{M}$ obtained from cost shaping with $\hat{V}^e$. As optimizing $\mathcal{M}$ is the same as optimizing the original MDP $\mathcal{M}_0$ (Ng et al., 1999), our formulation is equivalent to a pure RL approach on $\mathcal{M}_0$. In the extreme case when the oracle $\hat{V}^e$ has nothing to do with the true optimal oracle $V^*$, as there is no useful information we can distill from the oracle and RL becomes the only approach to solve $\mathcal{M}_0$.

## 5 EXPERIMENTS

We evaluated THOR on robotics simulators from OpenAI Gym (Brockman et al., 2016). Throughout this section, we report reward instead of cost, since OpenAI Gym by default uses reward. The baseline we compare against is TRPO-GAE (Schulman et al., 2016) and AGGREVATED (Sun et al., 2017).

To simulate oracles, we first train TRPO-GAE until convergence to obtain a policy as an expert $\pi^e$. We then collected a batch of trajectories by executing $\pi^e$. Finally, we use TD learning Sutton (1988) to train a value function $\hat{V}^e$ that approximates $V^e$. In all our experiments, we ignored $\pi^e$ and only used the pre-trained $\hat{V}^e$ for reward shaping. Hence our experimental setting simulates the situation where *we only have a batch of expert demonstrations available*, and not the experts themselves. This is a much harder setting than the interactive setting considered in previous work (Ross et al., 2011; Sun et al., 2017; Chang et al., 2015). Note that $\pi^e$ is not guaranteed to be an optimal policy, and $\hat{V}^e$ is only trained on the demonstrations from $\pi^e$, therefore the oracle $\hat{V}^e$ is just a coarse estimator of $V_{\mathcal{M}_0}^*$. Our goal is to show that, compared to AGGREVATED, THOR with $k > 1$ results in significantly better performance; compared to TRPO-GAE, THOR with some $k << H$ converges faster and is more sample efficient. For fair comparison to RL approaches, we do not pre-train policy or critic $\hat{A}$ using demonstration data, though initialization using demonstration data is suggested in theory and has been used in practice to boost the performance (Ross et al., 2011; Bahdanau et al., 2016).

For all methods we report statistics (mean and standard deviation) from 25 seeds that are i.i.d generated. For trust region optimization on the actor $\pi_\theta$ and GAE on the critic, we simply use the recommended parameters in the code-base from TRPO-GAE (Schulman et al., 2016). We did not tune any parameters except the truncation length $k$.

## 5.1 DISCRETE ACTION CONTROL

We consider two discrete action control tasks with sparse rewards: Mountain-Car, Acrobot and a modified sparse reward version of CartPole. All simulations have sparse reward in the sense that no reward signal is given until the policy succeeds (e.g., Acrobot swings up). In these settings, pure RL approaches that rely on random exploration strategies, suffer from the reward sparsity. On the other

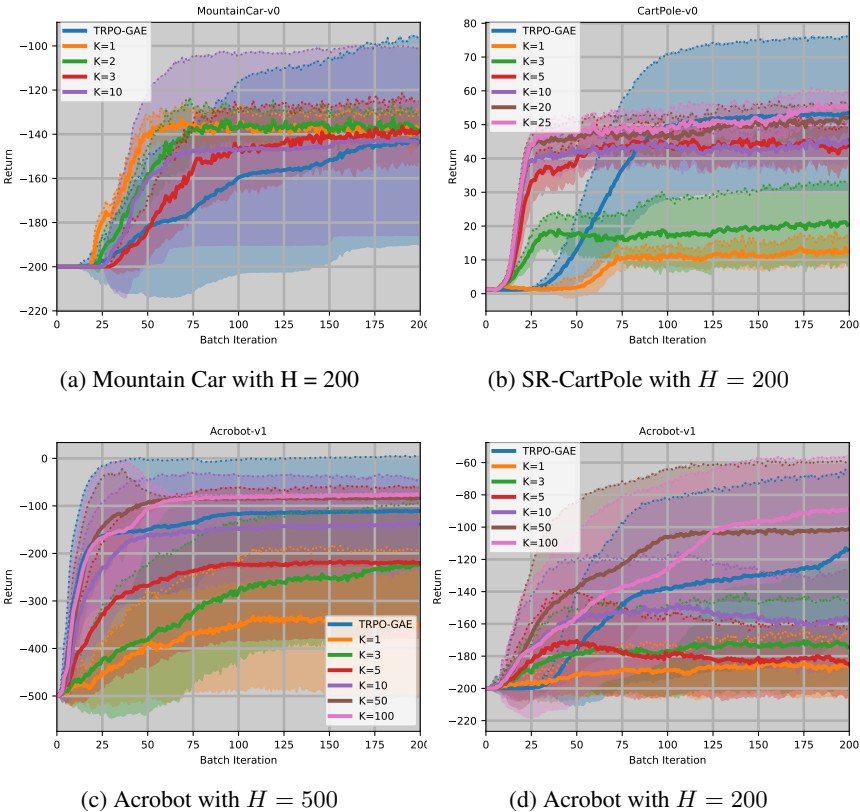

Figure 1: Reward versus batch iterations of THOR with different $k$ and TRPO-GAE (blue) for Mountain car, Sparse Reward (SR) CartPole, and Acrobot with different horizon. Average rewards across 25 runs are shown in solid lines and averages + std are shown in dotted lines.

hand, THOR can leverage oracle information for more efficient exploration. Results are shown in Fig. 1.

Note that in our setting where $\hat{V}^e$ is imperfect, THOR with $k > 1$ works much better than AGGREVATED (THOR with $k = 1$) in Acrobot. In Mountain Car, we observe that AGGREVATED achieves good performance in terms of the mean, but THOR with $k > 1$ (especially $k = 10$) results in much higher mean+std, which means that once THOR receives the reward signal, it can leverage this signal to perform better than the oracles.

We also show that THOR with $k > 1$ (but much smaller than $H$) can perform better than TRPO-GAE. In general, as $k$ increases, we get better performance. We make the acrobot setting even harder by setting $H = 200$ to even reduce the chance of a random policy to receive reward signals. Compare Fig. 1 (c) to Fig. 1 (b), we can see that THOR with different settings of $k$ always learns faster than TRPO-GAE, and THOR with $k = 50$ and $k = 100$ significantly outperform TRPO-GAE in both mean and mean+std. This indicates that THOR can leverage both reward signals (to perform better than AGGREVATED) and the oracles (to learn faster or even outperform TRPO).

## 5.2 CONTINUOUS ACTION CONTROL

We tested our approach on simulators with continuous state and actions from MuJoCo simulators: a modified sparse reward Inverted Pendulum, a modifed sparse reward Inverted Double Pendulum, Hopper and Swimmer. Note that, compared to the sparse reward setting, Hopper and Swimmer do not have reward sparsity and policy gradient methods have shown great results (Schulman et al., 2015; 2016). Also, due to the much larger and more complex state space and control space compared to the simulations we consider in the previous section, the value function estimator $\hat{V}^e$ is much less

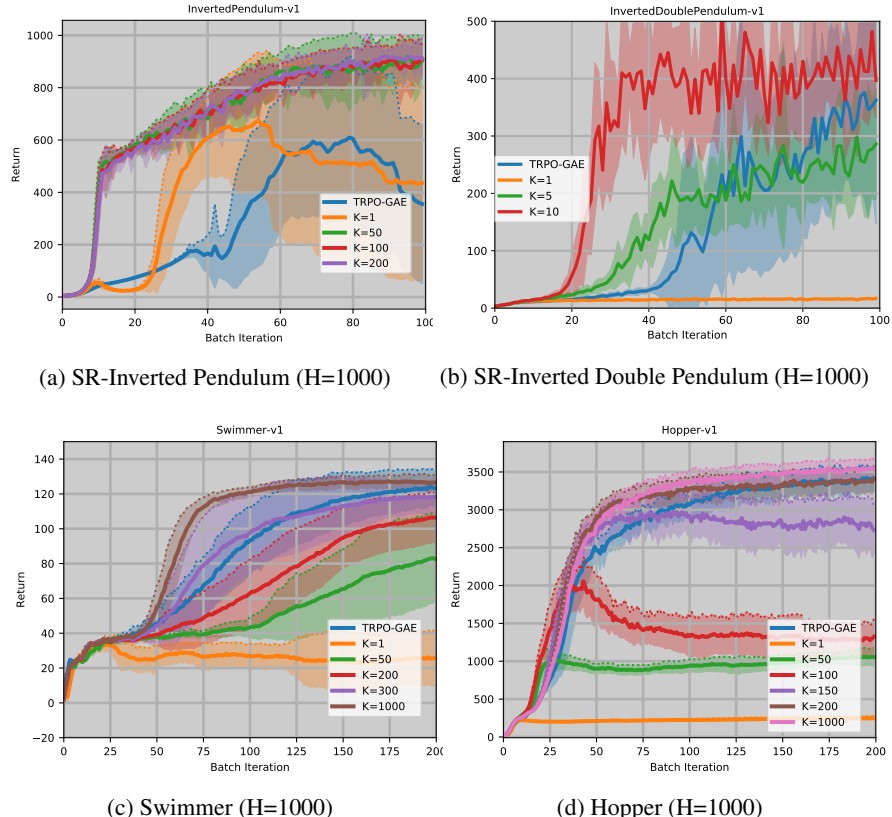

Figure 2: Reward versus batch iterations of THOR with different $k$ and TRPO-GAE (blue) for Sparse Reward (SR) Inverted Pendulum, Sparse Reward Inverted-Double Pendulum, Swimmer and Hopper. Average rewards across 25 runs are shown in solid lines and averages + std are shown in dotted lines.

accurate in terms of estimating $V^*_{\mathcal{M}_0}$ since the trajectories demonstrated from experts may only cover a very small part of the state and control space. Fig. 2 shows the results of our approach. For all simulations, we require $k$ to be around $20\% \sim 30\%$ of the original planning horizon $H$ to achieve good performance. AGGREVATED ($k = 1$) learned very little due to the imperfect value function estimator $\hat{V}^e$. We also tested $k = H$, where we observe that reward shaping with $\hat{V}^e$ gives better performance than TRPO-GAE. This empirical observation is consistent with the observation from (Ng et al., 1999) ( Ng et al. (1999) used SARSA (Sutton, 1988), not policy gradient based methods). This indicates that even when $\hat{V}^e$ is not close to $V^*$, policy gradient methods can still employ the oracle $\hat{V}^e$ just via reward shaping.

Finally, we also observed that our approach significantly reduces the variance of the performance of the learned polices (e.g., Swimmer in Fig. 2(a)) in *all* experiments, including the sparse reward setting. This is because truncation can significantly reduce the variance from the policy gradient estimation when $k$ is small compared to $H$.

## 6 CONCLUSION

We propose a novel way of combining IL and RL through the idea of cost shaping with an expert oracle. Our theory indicates that cost shaping with the oracle shortens the learner's planning horizon as a function of the accuracy of the oracle compared to the optimal policy's value function. Specifically, when the oracle is the optimal value function, we show that by setting $k = 1$ reveals previous imitation learning algorithm AGGREVATED. On the other hand, we show that when the oracle is imperfect, using planning horizon $k > 1$ can learn a policy that outperforms a policy that would been learned by AGGREVATE and AGGREVATED (i.e., $k = 1$). With this insight, we propose

THOR (Truncated HORizon policy search), a gradient based policy search algorithm that explicitly focusing on minimizing the total cost over a finite planning horizon. Our formulation provides a natural half-way point between IL and RL, and experimentally we demonstrate that with a reasonably accurate oracle, our approach can outperform RL and IL baselines.

We believe our high-level idea of shaping the cost with the oracle and then focusing on optimizing a shorter planning horizon is not limited to the practical algorithm we proposed in this work. In fact our idea can be combined with other RL techniques such as Deep Deterministic Policy Gradient (DDPG) (Lillicrap et al., 2015), which has an extra potential advantage of storing extra information from the expert such as the offline demonstrations in its replay buffer (Večerík et al. (2017)). Though in our experiments, we simply used some expert's demonstrations to pre-train $\hat{V}^e$ using TD learning, there are other possible ways to learn a more accurate $\hat{V}^e$. For instance, if an expert is available during training (Ross et al., 2011), one can online update $\hat{V}^e$ by querying expert's feedback.

## ACKNOWLEDGEMENT

Wen Sun is supported in part by Office of Naval Research contract N000141512365. The authors also thank Arun Venkatraman and Geoff Gordon for value discussion.

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

## A PROOF OF THEOREM 3.1

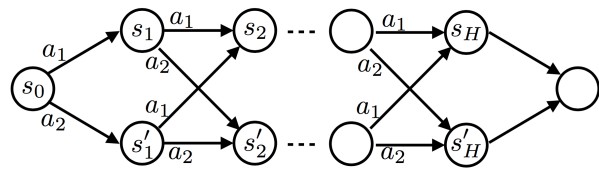

Figure 3: The special MDP we constructed for theorem 3.1

*Proof.* We prove the theorem by constructing a special MDP shown in Fig 3, where $H = \infty$. The MDP has deterministic transition, $2H + 2$ states, and each state has two actions $a_1$ and $a_2$ as shown in Fig. 3. Every episode starts at state $s_0$. For state $s_i$ (states on the top line), we have $c(s_i) = 0$ and for state $s'_i$ (states at the bottom line) we have $c(s_i) = 1$.

It is clear that for any state $s_i$, we have $Q^*(s_i, a_1) = 0$, $Q^*(s_i, a_2) = \gamma$, $Q^*(s'_i, a_1) = 1$ and $Q^*(s'_i, a_2) = 1 + \gamma$, for $i \geq 1$. Let us assume that we have an oracle $\hat{V}^e$ such that $\hat{V}^e(s_i) = 0.5 + \delta$ and $V^e(s'_i) = 0.5 - \delta$, for some positive real number $\delta$. Hence we can see that $|\hat{V}^e(s) - V^*(s)| = 0.5 + \delta$, for all $s$. Denote $\hat{Q}^e(s, a) = c(s, a) + \gamma\mathbb{E}_{s' \sim P_{sa}}[\hat{V}^e(s')]$, we know that $\hat{Q}^e(s_i, a_1) = \gamma(0.5 + \delta)$, $\hat{Q}^e(s_i, a_2) = \gamma(0.5 - \delta)$, $\hat{Q}^e(s'_i, a_1) = 1 + \gamma(0.5 + \delta)$ and $\hat{Q}^e(s'_i, a_2) = 1 + \gamma(0.5 - \delta)$.

It is clear that the optimal policy $\pi^*$ has cost $J(\pi^*) = 0$. Now let us compute the cost of the induced policy from oracle $\hat{Q}^e$: $\hat{\pi}(s) = \arg\min_a \hat{Q}^e(s, a)$. As we can see $\hat{\pi}$ makes a mistake at every state as $\arg\min_a \hat{Q}^e(s, a) \neq \arg\min_a Q^*(s, a)$. Hence we have $J(\hat{\pi}) = \frac{\gamma}{1-\gamma}$. Recall that in our constructed example, we have $\epsilon = 0.5 + \delta$. Now let $\delta \to 0^+$ (by $\delta \to 0^+$ we mean $\delta$ approaches to zero from the right side), we have $\epsilon \to 0.5$, hence $J(\hat{\pi}) = \frac{\gamma}{1-\gamma} \to \frac{2\gamma}{1-\gamma}\epsilon$ (due to the fact $2\epsilon \to 1$).

Hence we have $J(\hat{\pi}) - J(\pi^*) = \Omega\left(\frac{\gamma}{1-\gamma}\epsilon\right)$ $\qquad\qquad\square$

## B PROOF OF THEOREM 3.2

Below we prove Theorem 3.2.

*Proof of Theorem 3.2.* In this proof, for notation simplicity, we denote $V^\pi_{\mathcal{M}_0}$ as $V^\pi$ for any $\pi$. Using the definition of value function $V^\pi$, for any state $s_1 \in \mathcal{S}$ we have:

$$V^{\hat{\pi}^*}(s_1) - V^*(s_1)$$

$$= \mathbb{E}\left[\sum_{i=1}^{k}\gamma^{i-1}c(s_i, a_i) + \gamma^k V^{\hat{\pi}^*}(s_{k+1})|\hat{\pi}^*\right] - \mathbb{E}\left[\sum_{i=1}^{k}\gamma^{i-1}c(s_i, a_i) + \gamma^k V^*(s_{k+1})|\pi^*\right]$$

$$= \mathbb{E}\left[\sum_{i=1}^{k}\gamma^{i-1}c(s_i, a_i) + \gamma^k V^{\hat{\pi}^*}(s_{k+1})|\hat{\pi}^*\right] - \mathbb{E}\left[\sum_{i=1}^{k}\gamma^{i-1}c(s_i, a_i) + \gamma^k V^*(s_{k+1})|\hat{\pi}^*\right]$$

$$+ \mathbb{E}\left[\sum_{i=1}^{k}\gamma^{i-1}c(s_i, a_i) + \gamma^k V^*(s_{k+1})|\hat{\pi}^*\right] - \mathbb{E}\left[\sum_{i=1}^{k}\gamma^{i-1}c(s_i, a_i) + \gamma^k V^*(s_{k+1})|\pi^*\right]$$

$$= \gamma^k\mathbb{E}\left[V^{\hat{\pi}^*}(s_{k+1}) - V^*(s_{k+1})\right]$$

$$+ \mathbb{E}\left[\sum_{i=1}^{k}\gamma^{i-1}c(s_i, a_i) + \gamma^k V^*(s_{k+1})|\hat{\pi}^*\right] - \mathbb{E}\left[\sum_{i=1}^{k}\gamma^{i-1}c(s_i, a_i) + \gamma^k V^*(s_{k+1})|\pi^*\right]$$

$$(15)$$

Using the fact that $\|V^*(s) - \hat{V}^e(s)\| \le \epsilon$, we have that:

$$\mathbb{E}\Big[\sum_{i=1}^{k}\gamma^{i-1}c(s_i,a_i) + \gamma^k V^*(s_{k+1})|\hat{\pi}^*\Big] \le \mathbb{E}\Big[\sum_{i=1}^{k}\gamma^{i-1}c(s_i,a_i) + \gamma^k \hat{V}^e(s_{k+1})|\hat{\pi}^*\Big] + \gamma^k\epsilon,$$

$$\mathbb{E}\Big[\sum_{i=1}^{k}\gamma^{i-1}c(s_i,a_i) + \gamma^k V^*(s_{k+1})|\pi^*\Big] \ge \mathbb{E}\Big[\sum_{i=1}^{k}\gamma^{i-1}c(s_i,a_i) + \gamma^k \hat{V}^e(s_{k+1})|\pi^*\Big] - \gamma^k\epsilon.$$

Substitute the above two inequality into Eq. 15, we have:

$$V^{\hat{\pi}^*}(s_1) - V^*(s_1) \le \gamma^k\mathbb{E}\Big[V^{\hat{\pi}^*}(s_{k+1}) - V^*(s_{k+1})\Big] + 2\gamma^k\epsilon$$

$$+ \mathbb{E}\Big[\sum_{i=1}^{k}\gamma^{i-1}c(s_i,a_i) + \gamma^k\hat{V}^e(s_{k+1})|\hat{\pi}^*\Big] - \mathbb{E}\Big[\sum_{i=1}^{k}\gamma^{i-1}c(s_i,a_i) + \gamma^k\hat{V}^e(s_{k+1})|\pi^*\Big]$$

$$\le \gamma^k\mathbb{E}\Big[V^{\hat{\pi}^*}(s_{k+1}) - V^*(s_{k+1})\Big] + 2\gamma^k\epsilon$$

$$+ \mathbb{E}\Big[\sum_{i=1}^{k}\gamma^{i-1}c(s_i,a_i) + \gamma^k\hat{V}^e(s_{k+1})|\hat{\pi}^*\Big] - \mathbb{E}\Big[\sum_{i=1}^{k}\gamma^{i-1}c(s_i,a_i) + \gamma^k\hat{V}^e(s_{k+1})|\hat{\pi}^*\Big]$$

$$= \gamma^k\mathbb{E}\Big[V^{\hat{\pi}^*}(s_{k+1}) - V^*(s_{k+1})\Big] + 2\gamma^k\epsilon, \tag{16}$$

where the second inequality comes from the fact that $\hat{\pi}^*$ is the minimizer from Eq. 6. Recursively expand $V^{\pi^*}(s_{k+1}) - V^*(s_{k+1})$ using the above procedure, we can get:

$$V^{\hat{\pi}^*}(s_1) - V^*(s_1) \le 2\gamma^k\epsilon(1 + \gamma^k + \gamma^{2k} + ...) \le \frac{2\gamma^k}{1-\gamma^k}\epsilon = O(\frac{\gamma^k}{1-\gamma^k}\epsilon). \tag{17}$$

Since the above inequality holds for any $s_1 \in \mathcal{S}$, for any initial distribution $v$ over state space $\mathcal{S}$, we will have $J(\hat{\pi}^*) - J(\pi^*) \le O(\frac{\gamma^k}{1-\gamma^k}\epsilon)$. $\qquad\square$

