# OpenReview forum: "TRUNCATED HORIZON POLICY SEARCH: COMBINING REINFORCEMENT LEARNING & IMITATION LEARNING"
_ICLR.cc/2018/Conference — Accept (Poster)_

### Official Review · AnonReviewer2 · 2017-11-27
**Lack of rigor and claim (that k=1 does not allow the RL agent to outperform the expert) not justified**

**Rating:** 3
**Confidence:** 5

**Review:**

This work proposes to use the value function V^e of some expert policy \pi^e in order to speed up learning of an RL agent which should eventually do better than the expert. The emphasis is put on using k-steps (with k>1) Bellman updates using bootstrapping from V^e.

It is claimed that the case k=1 does not allow the agent to outperform the expert policy, whereas k>1 does (Section 3.1, paragraph before Lemma 3.2).

I disagree with this claim. Indeed a policy gradient algorithm (similar to (10)) with a 1-step advantage c(s,a) + gamma V^e(s_{t+1}) - V^e(s_t) will converge (say in the tabular case, or in the case you consider of a rich enough policy space \Pi) to the greedy policy with respect to V^e, which is strictly better than V^e (if V^e is not optimal). So you don’t need to use k>1 to improve the expert policy. Now it’s true that this will not converge to the optimal policy (since you keep bootstrapping with V^e instead of the current value function), but neither the k-step advantage will.

So I don’t see any fundamental difference between k=1 and k>1. The only difference being that the k-step bootstrapping will implement a k-step Bellman operator which contracts faster (as gamma^k) when k is large. But the best choice of k has to be discussed in light of a bias-variance discussion, which is missing here. So I find that the main motivation for this work is not well supported.

Algorithmic suggestion:
Instead of bootstrapping with V^e, why not bootstrap with min(V^e, V), where V is your current approximation of the value function. In that way you would benefit from (1) fast initialization with V^e at the beginning of learning, (2) continual improvement once you’ve reached the performance of the expert.

Other comments:
Requiring that we know the value function of the expert on the whole state space is a very strong assumption that we do not usually make in Imitation learning. Instead we assume we have trajectories from expert (from which we can compute value function along those trajectories only). Generalization of the value function to other states is a hard problem in RL and is the topic of important research.

The overall writing lacks rigor and the contribution is poor. Indeed the lower bound (Theorem 3.1) is not novel (btw, the constant hidden in the \Omega notation is 1/(1-gamma)). Theorems 3.2 and 3.3 are not novel either. Please read [Bertsekas and Tsitsiklis, 96] as an introduction to dynamic programming with approximation.

The writing could be improved, and there are many typos, such as:
- J is not defined (Equation (2))
- Why do you call A a disadvantage function whereas this quantity is usually called an advantage?
- You are considering a finite (ie, k) horizon setting, so the value function depend on time. For example the value functions defined in (11) depend on time.
- All derivations in Section 4, before subsection 4.1 are very approximate and lack rigor.
- Last sentence of Proof of theorem 3.1. I don’t understand H -> 2H epsilon. H is fixed, right? Also your example does not seem to be a discounted problem.

---

> ### Author Response · Authors · 2018-01-01
> **RE: Lack of rigor and claim (that k=1 does not allow the RL agent to outperform the expert) not justified**
>
> We thank the reviewer for constructive feedback, below R stands for Reviewer and A stands for our answer
>
> R: K=1 VS K > 1:
>
> A: We do not agree with the reviewer on this point. First, when k=infinity, we can find the optimal policy (under the assumption that we can fully optimize the MDP, of course) which will be significantly better than an imperfect expert pi^e. While we agree that when K=1, the greedy policy with respect to V^e can outperform the expert, the greedy policy is only a *one-step* deviation improvement of V^e. If pi^e is far away from optimality, the one-step improvement greedy policy will likely be far away as well. This is shown in the lower bound analysis.  Our main theorem clearly shows that as k increases, the learned policy is getting closer and closer to optimality. Combine the lower bound on the performance of the one-step greedy policy, and the upper bound on the performance of the learned policy with k>1, and it is clear that the learned policy under k>1 is closer to optimality.  In summary, we are emphasizing that with k>1, we can learn a policy that is even better than the greedy policy with respect Q^e (not just pi^e), which is the best one can learn if one uses previous algorithm such as AggreVaTe.
>
> We have clarified this point in the revised version of the paper.
>
> R: Regarding bias-variance tradeoff
>
> Unlike previous work (e.g., TD learning with k-step look ahead), we are using V^e as the termination value instead of V^pi. The main argument of our work is that by reasoning k steps into the future with V^e as the termination value, we are trading between learning complexity and the optimality. When k = 1, we are greedy with respect to the one-step cost, but the downside is that we can only hope that the learned policy is, at best, a one-step deviation improvement over the expert. When k > 1, but less than infinity, we are essentially solving a MDP that is easier than the original MDP due to the shorter horizon, but with the benefit of learning a policy that is moving closer to the optimality than the greedy policy with respect to Q^e. In our theorem, gamma^k does not merely serve as a contraction factor as it did in, for example, TD learning. gamma^k here serves as a measure on how close the learned policy is to the optimal policy.
>
> R: “Requiring know the value function of the expert on the whole state space is a very strong assumption...”
>
> We agree with the reviewer. In our experiments, we learned V^e from a set of demonstrations and then used the learned V^e. As we showed, previous work AggreVaTe(D) performs poorly in this setting, as V^e may only be an accurate oracle in the state space of the expert’s demonstrations. For challenging problems with very large state action spaces, we need to assume that \pi^e exists so that V^e can be estimated by rollouts (the same assumption DAgger used (Ross & Bagnell 11, AISTATS)). Luckily, this kind of oracles do exist in practice via access to a simulator, a search algorithm, even in real robotics applications (e.g., see Choudhury et.al, 17, ICRA, Pan et.al, 17), and in natural language processing, where the ground-truth label information can be leveraged to construct experts (Chang et al, 15, ICML, Sun et al, 17 ICML).  In these applications, we cannot guarantee the constructed “expert” is globally optimal, hence our results can be directly applied.
>
> R: "J is not defined..":
>
> We thank the reviewer for pointing this out. We have revised the draft accordingly.
>
> R: proof of theorem 3.1
>
> We thank the reviewer for pointing out the confusion. Although the general proof strategy is not changed, we have revised the proof and also changed it to the setting with discount factor and infinite horizon to make it consistent with the main setting in the paper.

---

### Official Review · AnonReviewer1 · 2017-11-28
**The main idea has been studied before, but not properly discussed.**

**Rating:** 6
**Confidence:** 4

**Review:**

=== SUMMARY ===

The paper considers a combination of Reinforcement Learning (RL) and Imitation Learning (IL), in the infinite horizon discounted MDP setting.
The IL part is in the form of an oracle that returns a value function V^e, which is an approximation of the optimal value function. The paper defines a new cost (or reward) function based on V^e, through shaping (Eq. 1). It is known that shaping does not change the optimal policy.

A key aspect of this paper is to consider a truncated horizon problem (say horizon k) with the reshaped cost function, instead of an infinite horizon MDP.
For this truncated problem, one can write the (dis)advantage function as a k-step sum of reward plus the value returned by the oracle at the k-th step (cf. Eq. 5).
Theorem 3.3 shows that the value of the optimal policy of the truncated MDP w.r.t. the original MDP is only O(gamma^k eps) worse than the optimal policy of the original problem (gamma is the discount factor and eps is the error between V^e and V*).

This suggests two things:
1) Having an oracle that is accurate (small eps) leads to good performance. If oracle is the same as the optimal value function, we do not need to plan more than a single step ahead.
2) By planning for k steps ahead, one can decrease the error in the oracle geometrically fast. In the limit of k —> inf, the error in the oracle does not matter.

Based on this insight, the paper suggests an actor-critic-like algorithm called THOR (Truncated HORizon policy search) that minimizes the total cost over a truncated horizon with a modified cost function.

Through a series of experiments on several benchmark problems (inverted pendulum, swimmer, etc.), the paper shows the effect of planning horizon k.



=== EVALUATION & COMMENTS ===

I like the main idea of this paper. The paper is also well-written. But one of the main ideas of this paper (truncating the planning horizon and replacing it with approximation of the optimal value function) is not new and has been studied before, but has not been properly cited and discussed.

There are a few papers that discuss truncated planning. Most closely is the following paper:

Farahmand, Nikovski, Igarashi, and Konaka, “Truncated Approximate Dynamic Programming With Task-Dependent Terminal Value,” AAAI, 2016.

The motivation of AAAI 2016 paper is different from this work. The goal there is to speedup the computation of finite, but large, horizon problem with a truncated horizon planning. The setting there is not the combination of RL and IL, but multi-task RL. An approximation of optimal value function for each task is learned off-line and then used as the terminal cost.
The important point is that the learned function there plays the same role as the value provided by the oracle V^e in this work. They both are used to shorten the planning horizon. That paper theoretically shows the effect of various error terms, including terms related to the approximation in the planning process (this paper does not do that).

Nonetheless, the resulting algorithms are quite different. The result of this work is an actor-critic type of algorithm. AAAI 2016 paper is an approximate dynamic programming type of algorithm.

There are some other papers that have ideas similar to this work in relation to truncating the horizon. For example, the multi-step lookahead policies and the use of approximate value function as the terminal cost in the following paper:

Bertsekas, “Dynamic Programming and Suboptimal Control: A Survey from ADP to MPC,” European Journal of Control, 2005.

The use of learned value function to truncate the rollout trajectory in a classification-based approximate policy iteration method has been studied by

Gabillon, Lazaric, Ghavamzadeh, and Scherrer, “Classification-based Policy Iteration with a Critic,” ICML, 2011.

Or in the context of Monte Carlo Tree Search planning, the following paper is relevant:

Silver et al., “Mastering the game of Go with deep neural networks and tree search,” Nature, 2016.

Their “value network” has a similar role to V^e. It provides an estimate of the states at the truncated horizon to shorten the planning depth.

Note that even though these aforementioned papers are not about IL, this paper’s stringent requirement of having access to V^e essentially make it similar to those papers.


In short, a significant part of this work’s novelty has been explored before. Even though not being completely novel is totally acceptable, it is important that the paper better position itself compared to the prior art.


Aside this main issue, there are some other comments:


- Theorem 3.1 is not stated clearly and may suggest more than what is actually shown in the proof. The problem is that it is not clear about the fact the choice of eps is not arbitrary.
The proof works only for eps that is larger than 0.5. With the construction of the proof, if eps is smaller than 0.5, there would not be any error, i.e., J(\hat{pi}^*) = J(pi^*).

The theorem basically states that if the error is very large (half of the range of value function), the agent does not not perform well. Is this an interesting case?


- In addition to the papers I mentioned earlier, there are some results suggesting that shorter horizons might be beneficial and/or sufficient under certain conditions. A related work is a theorem in the PhD dissertation of Ng:

Andrew Ng, Shaping and Policy Search in Reinforcement Learning, PhD Dissertation, 2003.
(Theorem 5 in Appendix 3.B: Learning with a smaller horizon).

It is shown that if the error between Phi (equivalent to V^e here) and V* is small, one may choose a discount factor gamma’ that is smaller than gamma of the original MDP, and still have some guarantees. As the discount factor has an interpretation of the effective planning horizon, this result is relevant. The result, however, is not directly comparable to this work as the planning horizon appears implicitly in the form of 1/(1-gamma’) instead of k, but I believe it is worth to mention and possibly compare.

- The IL setting in this work is that an oracle provides V^e, which is the same as (Ross & Bagnell, 2014). I believe this setting is relatively restrictive as in many problems we only have access to (state, action) pairs, or sequence thereof, and not the associated value function. For example, if a human is showing how a robot or a car should move, we do not easily have access to V^e (unless the reward function is known and we estimate the value with rollouts; which requires us having a long trajectory). This is not a deal breaker, and I would not consider this as a weakness of the work, but the paper should be more clear and upfront about this.


- The use of differential operator nabla instead of gradient of a function (a vector field) in Equations (10), (14), (15) is non-standard.

- Figures are difficult to read, as the colors corresponding to confidence regions of different curves are all mixed up. Maybe it is better to use standard error instead of standard deviation.


===
After Rebuttal: Thank you for your answer. The revised paper has been improved. I increase my score accordingly.

---

> ### Author Response · Authors · 2018-01-01
> **RE: The main idea has been studied before, but not properly discussed.**
>
> We thank the reviewer for constructive feedback, below R stands for Reviewer and A stands for our answer
>
> R: “Regarding to some previous work using truncated horizon”:
>
> A: we gratefully thank the reviewer for pointing out all of these related previous works. We agree with the reviewer that the idea of using truncated horizon has been explored before. Following the suggestions from the reviewer, we have revised the paper to better position our work. Please see Sec 1.1 in the revision for a discussion on related work and our contributions.
>
> Some of the previous work that uses the idea of truncating horizon mainly focuses on using V^{\pi} as the termination value for bias-variance tradeoff.  We used V^e, instead of V^{\pi}. We think using V^{e} with truncated horizon allows us to interpolate between pure IL and full RL and trades between sample complexity and the optimality: with k=1, the reshaped MDP is easy to solve as it's one-step greedy and we reveal IL algorithm AggreVaTeD, but at the cost of only learning a policy that has similar performance as the expert. When k > 1, we face a MDP that is between the one-step greedy MDP and the original full horizon MDP. Solving a truncated horizon MDP with k > 1 is harder than the one-step greedy MDP, but at the benefit of outperforming the expert and getting closer to optimality. Another contribution of the paper is an efficient actor-critic like algorithm for continuous MDPs, which is not available in previous work (e.g., Ng's thesis work)
>
> R: "Theorem 5 in Appendix 3.B in Ng's PhD dissertation":
>
> A: Again, we gratefully thank the reviewer for pointing out this theorem that we were not aware of!  We agree with the reviewer, a smaller discount factor in theory is equivalent to a shorter planning horizon. One of the advantages of explicitly using a truncated horizon is for reducing computational complexity, as mentioned by some other previous work.  Though we agree that our main theorem is similar to Theorem 5 in the appendix of Ng's dissertation,  we would like to emphasize that one of the contributions of our work is that we interpret expert's value function as a potential function and this could help us explain and generalize previous imitation learning and bridge the gap between IL and RL.
>
> R: the choice of \eps in the proof of theorem 3.1
>
> A: We believe we can construct similar MDPs to make \eps smaller by increasing the number of actions. We believe we can make \eps small around 1/|A|, where |A| is the number of actions. The main idea is that we want to construct a MDP where for each state s, the value A^e(s,a) itself is close to the value A*(s,a), but the order on actions induced by A^e is different from the order of actions induced by A*(s,a), forcing the greedy policy with respect to A^e to make mistakes.

---

### Official Review · AnonReviewer3 · 2017-11-28

**Rating:** 7
**Confidence:** 3

**Review:**

This paper proposes a new theoretically-motivated method for combining reinforcement learning and imitation learning for acquiring policies that are as good as or superior to the expert. The method assumes access to an expert value function (which could be trained using expert roll-outs) and uses the value function to shape the reward function and allow for truncated-horizon policy search. The algorithm can gracefully handle suboptimal demonstrations/value functions, since the demonstrations are only used for reward shaping, and the experiments demonstrate faster convergence and better performance compared to RL and AggreVaTeD on a range of simulated control domains. The paper is well-written and easy to understand.

My main feedback is with regard to the experiments:
I appreciate that the experiments used 25 random seeds! This provides a convincing evaluation.
It would be nice to see experimental results on even higher dimensional domains such as the ant, humanoid, or vision-based tasks, since the experiments seem to suggest that the benefit of the proposed method is diminished in the swimmer and hopper domains compared to the simpler settings.
Since the method uses demonstrations, it would be nice to see three additional comparisons: (a) training with supervised learning on the expert roll-outs, (b) initializing THOR and AggreVaTeD (k=1) with a policy trained with supervised learning, and (c) initializing TRPO with a policy trained with supervised learning. There doesn't seem to be any reason not to initialize in such a way, when expert demonstrations are available, and such an initialization should likely provide a significant speed boost in training for all methods.
How many demonstrations were used for training the value function in each domain? I did not see this information in the paper.

With regard to the method and discussion:
The paper discusses the connection between the proposed method and short-horizon imitation and long-horizon RL, describing the method as a midway point. It would also be interesting to see a discussion of the relation to inverse RL, which considers long-term outcomes from expert demonstrations. For example, MacGlashn & Littman propose a midway point between imitation and inverse RL [1].
Theoretically, would it make sense to anneal k from small to large? (to learn the most effectively from the smallest amount of experience)

[1] https://www.ijcai.org/Proceedings/15/Papers/519.pdf


Minor feedback:
- The RHS of the first inequality in the proof of Thm 3.3 seems to have an error in the indexing of i and exponent, which differs from the line before and line after

**Edit after rebuttal**: I have read the other reviews and the authors' responses. My score remains the same.

---

> ### Author Response · Authors · 2018-01-01
> **Re:Review**
>
> We thank the reviewer for constructive feedback. Below R stands for Reviewer and A stands for our answers.
>
> R: “Experiment on higher-dimensional domains”:
>
> A: In our experiments, we investigated the option of using a V^e pre-trained from expert demonstrations. In higher-dimensional tasks, training a globally accurate value function purely from a batch of expert demonstrations is difficult due to the large state space and the fact that the expert demonstrations only cover a tiny part of the state space. This is the main reason we think the Swimmer and Hopper experiment did not show the clear advantage of our approach. For higher-dimensional tasks, we believe we need a stronger assumption on the availability of the expert. For example, we may require oracles to exist in the training loop (similar to the assumptions used in previous IL work such as DAgger (Ross et.al, 2011)) so that we can estimate and query V^e(s) on the fly. This kind of oracle can exist in practice: for example access to a simulator, a search algorithm, and in real robotics applications (e.g., see Choudhury et.al, 17, ICRA, Pan et.al, 17, where optimal controllers were used as oracles), but also in natural language processing, where the ground truth label information can be leveraged to construct experts (Chang et al, 15, ICML, Sun et al, 17 ICML).  In these applications, we can not guarantee the constructed “expert” is globally optimal, hence we believe our work can be directly applied and result in improved performance. Experiments like these are left to future work.
>
> R: Compare to Supervised Training:
>
> A: We thank the reviewer for this suggestion, we are working on this and will include a comparison to a simple supervised learning approach, although previous work has demonstrated that supervised learning won’t work as well as the interactive learning setting (Ross & Bagnell, 2011, AISTATS) in both theory and in practice.
>
> R: “how many demonstrations:”
>
> A: We used a number of demonstration trajectories ranging from 10 to 100 (almost the same as the number of trajectories we use in each batch during learning), depending on the task. For higher-dimensional tasks, a larger number of state-action pairs collected from demonstrations are needed. This is simply due to the fact that the feature space has a higher dimension.
>
> R: “Regarding to previous work on Imitation learning and inverse RL”
>
> A: We thank the reviewer for pointing out this paper. This paper is related and this paper also shows the advantage of using a truncated horizon (e.g., less computational complexity), although the context is different: interpolating between behaviour cloning and inverse RL (or “intention learning,” in the paper’s words) by learning a reward functions and then planning with the known dynamics. Our work interpolates between a different imitation learning approach---interactive IL and RL. We think this work and our work together show the advantage of using a truncated horizon: less computational complexity (what this paper showed), and better performance than an imperfect expert (what we showed in our work).

---

> > ### Comment · AnonReviewer3 · 2018-01-10
> > **Response**
> >
> > Thank you for the response.
> >
> > Regarding experiments on higher-dimensional domains, I think the paper would benefit from adding this discussion, aiding the reader in understanding this potential limitation.
> >
> > Information about the number of demonstrations should also be added to the paper.

---

### Author Response · Authors · 2018-01-05
**Summary of the changes we made to the paper**

We thank the reviewers for constructive feedback and we have revised our paper based on the suggestions from the reviewers. Below we summarize the main changes we made to the paper:

1. We added a new paragraph at the end of the introduction section to summarize the related work and our contributions. We hope this will better position our work.

2. In Introduction, we clarified that previous imitation learning approach---AggreVaTe, can outperform an imperfect expert as well, but with only one step deviation improvement. We also emphasized by how much our approach can improve over AggreVaTe at the end of Theorem 3.2.

3. We updated Theorem 3.1 and Theorem 3.2 to include the dependency on 1/(1-\gamma) in the big O notation.

4. We updated the proof in Appendix so that it works for discounted and infinite horizon MDP,  though the main strategy is not changed. We also added a small paragraph at the end of Theorem 3.1 to illustrate the high-level strategy of the proof.

---

### Decision · Program_Chairs · 2018-01-29
**ICLR 2018 Conference Acceptance Decision**

**Decision:**

Accept (Poster)

**Comment:**

This paper proposes a theoretically-motivated method for combining reinforcement learning and imitation learning. There was some disagreement amongst the reviewers, but the AC was satisfied with the authors' rebuttal.